# Synergistic Effect of Treatment with Highly Pathogenic Porcine Reproductive and Respiratory Syndrome Virus and Lipopolysaccharide on the Inflammatory Response of Porcine Pulmonary Microvascular Endothelial Cells

**DOI:** 10.3390/v15071523

**Published:** 2023-07-08

**Authors:** Xinyue Yao, Wanwan Dai, Siyu Yang, Zhaoli Wang, Qian Zhang, Qinghui Meng, Tao Zhang

**Affiliations:** 1Beijing Key Laboratory of Traditional Chinese Veterinary Medicine, Animal Science and Technology College, Beijing University of Agriculture, No. 7 Beinong Road, Beijing 102206, Chinazhangqianbua@163.com (Q.Z.); 2College of Veterinary Medicine, Shanxi Agriculture University, Taigu 030801, China; 3Beijing Milu Ecological Research Center, Beijing Research Institute of Science and Technology, Beijing 100076, China

**Keywords:** porcine reproductive and respiratory syndrome virus, secondary infection, lipopolysaccharide, microvascular endothelial cells, inflammation, endothelial barrier, transendothelial cell migration

## Abstract

The highly pathogenic porcine reproductive and respiratory syndrome virus (HP-PRRSV) often causes secondary bacterial infection in piglets, resulting in inflammatory lung injury and leading to high mortality rates and significant economic losses in the pig industry. Microvascular endothelial cells (MVECs) play a crucial role in the inflammatory response. Previous studies have shown that HP-PRRSV can infect porcine pulmonary MVECs and damage the endothelial glycocalyx. To further understand the role of pulmonary MVECs in the pathogenesis of HP-PRRSV and its secondary bacterial infection, in this study, cultured porcine pulmonary MVECs were stimulated with a HP-PRRSV HN strain and lipopolysaccharide (LPS). The changes in gene expression profiles were analyzed through transcriptome sequencing, and the differentially expressed genes were verified using qRT-PCR, Western blot, and ELISA. Furthermore, the effects on endothelial barrier function and regulation of neutrophil trans-endothelial migration were detected using the Transwell model. HP-PRRSV primarily induced differential expression of numerous genes associated with immune response, including IFIT2, IFIT3, VCAM1, ITGB4, and CCL5, whereas LPS triggered an inflammatory response involving IL6, IL16, CXCL8, CXCL14, and ITGA7. Compared to the individual effect of LPS, when given after HN-induced stimulation, it caused a greater number of changes in inflammatory molecules, such as VCAM1, IL1A, IL6, IL16, IL17D, CCL5, ITGAV, IGTB8, and TNFAIP3A, a more significant reduction in transendothelial electrical resistance, and higher increase in neutrophil transendothelial migration. In summary, these results suggest a synergistic effect of HP-PRRSV and LPS on the inflammatory response of porcine pulmonary MVECs. This study provides insights into the mechanism of severe lung injury caused by secondary bacterial infection following HP-PRRSV infection from the perspective of MVECs, emphasizing the vital role of pulmonary MVECs in HP-PRRSV infection.

## 1. Introduction

Porcine reproductive and respiratory syndrome virus (PRRSV) is an enveloped, positive-sense RNA virus belonging to the family Arteriviridae [1]. The virus has undergone rapid genetic evolution, and in 2006, a highly pathogenic strain of PRRSV (HP-PRRSV) emerged in Jiangxi Province, China [2]. HP-PRRSV is characterized by more severe symptoms, higher morbidity, and increased mortality compared to PRRSV [3]. Notably, HP-PRRSV-infected pigs are often complicated by secondary bacterial infections, which are believed to contribute to the occurrence of severe clinical symptoms and pathological damage such as high fever and acute lung injury. Gram-negative bacteria, such as *Escherichia coli*, *Haemophilus parasuis*, *Vibrio cholerae*, *Bacillus bronchisepticum*, and *Pasteurella multocida*, frequently cause secondary infections of HP-PRRSV and can release high concentrations of lipopolysaccharide (LPS) in the piglet lungs [4,5], and even exist in dust within pig barns at various concentrations [6]. Qiao et al. investigated the secretion of pro-inflammatory cytokines in pulmonary alveolar macrophages (PAMs) stimulated with HP-PRRSV and LPS. Their results showed that HP-PRRSV induced the secretion of inflammatory cytokines, such as IL-1β and TNF-α, and the additional presence of LPS further increased their secretion. Moreover, the increase in cytokine secretion was more significant with a stronger virulent PRRSV strain [7]. Similarly, when pigs were inoculated with LPS and PRRSV in the trachea, they exhibited more severe respiratory symptoms compared to those inoculated with PRRSV alone [8].

Although PAMs are considered the main target cells for PRRSV and play a crucial role in maintaining lung homeostasis and defense, they alone cannot fully explain the pathogenesis of PRRSV. The lung, as a highly vascularized organ, contains massive microvascular endothelial cells (MVECs), which not only serve as the physical basis for gas–blood exchange but also act as a barrier against pathogens circulating through the bloodstream to reach the lungs or disseminate from the lung tissue to the rest of the body. Furthermore, MVECs can interact with lung resident cells, such as PAMs, and then regulate the recruitment and exudation of neutrophils and other immune cells. Consequently, MVECs are a key control point in various pathological changes, such as inflammation and fever [9], and their involvement in the pathogenesis of many viruses and bacterial toxins has been well documented [10,11].

Previous studies in our laboratory have shown that porcine pulmonary MVECs are susceptible to HP-PRRSV [12], and that HP-PRRSV infection damages the structure and function of the endothelial surface glycocalyx [13]. Additionally, MVECs exhibit a certain non-specific immune response to HP-PRRSV infection [14]. However, the functional changes in porcine pulmonary MVECs during secondary bacterial infection following HP-PRRSV infection remain unclear.

This experiment aimed to further understand the role of MVECs in severe lung injury caused by HP-PRRSV-mediated secondary bacterial infection. Porcine pulmonary MVECs were cultured in vitro and stimulated with HP-PRRSV HN strain and/or LPS. We analyzed the differential gene expression using transcriptome sequencing technology and validated the differential expression of partial inflammatory molecules at the mRNA and protein levels. We also examined changes in endothelial barrier function and the transendothelial migration (TEM) of neutrophils using the Transwell model. The results of this study explain the severe inflammatory damage in the lung caused by HP-PRRSV-mediated secondary bacterial infection and consolidate the key role of porcine pulmonary MVECs in the pathogenesis of HP-PRRSV.

## 2. Materials and Methods

### 2.1. Virus

The HP-PRRSV HN strain used in the experiment was generously donated by Dr. Zhanzhong Zhao from the Beijing Institute of Animal Husbandry and Veterinary Medicine, Chinese Academy of Agricultural Sciences. The virus was cultured and propagated with Marc-145 cells, and its titer was determined to be 10^−6.339^ TCID_50_/0.1 mL. Normal Marc-145 cells were treated using the same method and the cell lysate was collected as a control. Both the virus lysate and the control lysate were stored at −80 °C.

### 2.2. Cells

Porcine pulmonary MVECs were isolated from approximately 15-day-old SPF British Large White pigs (purchased from Beijing SPF Pig Breeding Management Center). All procedures were approved by the Animal Protection Committee of Beijing Agricultural University (No. BUA_ZT2022012) and performed according to the previous method with some modifications [15]. Briefly, the lung edge tissue was taken, minced after removal of the pleura, digested with 0.2% type II collagenase solution (Worthington, LS004176, Lakewood, NJ, USA), and dissociated into a single cell suspension, which was plated and incubated. The primary culture was preliminarily purified using the differential attachment method. Then, CD31-positive cells were separated using a high-throughput magnetic bead sorter (Thermo Fisher Scientific, Vantaa, Finland), and the purified MVECs were cultured in endothelial medium (M&C gene technology, L2305005, Beijing, China) containing 5% serum (Aoqing Biotech, AQmv09900, Beijing, China) and 0.25% ECGS (M&C gene technology, CC019, Beijing, China). Immunofluorescence staining for factor VIII was carried out for further identification, and cells from passage 3–5 were used for experiments.

Neutrophils were isolated from the peripheral blood of adult pigs obtained from the Beijing Fifth Meat Joint Processing Plant using the Neutrophil Isolation Kit (Solarbio, P4140, Beijing, China) following the provided instructions. The isolated neutrophils were adjusted to a density of 1 × 10^8^ cells/mL, stored at −80 °C, and used within one month. Neutrophils were recovered and resuspended in RPMI 1640 medium (Aoqing Biotech, AQ11875, Beijing, China) containing 5% serum at 1 × 10^8^ cells/mL, and cell viability was determined by staining with trypan blue solution (Solarbio, T8070, Beijing, China) before being used for experiments.

### 2.3. Transcriptome Sequencing

Porcine pulmonary MVECs grown to confluence in 10 cm Petri dishes were divided into four groups: control group, PRRSV group, LPS group, and PRRSV-LPS group, and each group had three replicates. In the PRRSV group and PRRSV-LPS group, cells were exposed to 3 mL of HP-PRRSV HN lysate for 1 h, and the control group and the LPS group were exposed to an equal amount of the control lysate for 1 h. After washing with D-Hank’s solution and incubated in a maintenance medium containing 2% serum for 36 h, the LPS group and the PRRSV-LPS group were treated with LPS at a final concentration of 5 μg/mL (Solarbio, L8880, Beijing, China), and the other two groups were treated with an equal volume of maintenance medium. Over another 24 h incubation, total cellular RNA was extracted from each group. Transcriptome sequencing analysis was carried out by Wekemo Tech Group Co., Ltd. (Shenzhen, China)and the procedure is briefly described as follows.

The RNA concentration and integrity were determined using Nanodrop and an Agilent 2100 bioanalyzer, and enriched using magnetic beads connected with oligo-thymine to capture the polyadenylated tail. The enriched mRNA was then fragmented randomly using divalent cations in NEB Fragmentation Buffer, followed by NEB general library construction. The library was preliminarily quantified using a Qubit 2.0 Fluorometer, and the fragment length was assessed using an Agilent 2100 bioanalyzer. Accurate quantification of the library’s effective concentration was performed using qRT-PCR to ensure a concentration higher than 2 nM. High-throughput sequencing was carried out using the Illumina system. After filtering the raw data, the sequences were aligned to the pig genome “https://ftp.ncbi.nlm.nih.gov/genomes/all/GCF/000/003/025/GCF_000003025.6_Sscrofa11.1/GCF_000003025.6_Sscrofa11.1_genomic.fna.gz (accessed on 4 January 2022)” using HISAT2 software. The aligned sequences were then assembled and quantified using String Tie. To identify differentially expressed genes, the threshold criteria of Fold Change > 1.5 and adjusted *p*-value (Padj) < 0.05 were applied. Functional enrichment analysis of the differentially expressed gene sets was performed using GO functional enrichment analysis and KEGG pathway enrichment analysis with cluster Profiler 3.14.3 software.

### 2.4. Primer Design and qRT-PCR

To validate the results from RNA sequencing, qRT-PCR was performed to detect the expression of eight differentially expressed genes related to the inflammatory response. The primer sequences were obtained from the data on the “https://www.ncbi.nlm.nih.gov (accessed on 1 June 2022)”website (Table 1) and were synthesized by Sangon Biotech (Shanghai) Co., Ltd. The total cellular RNA extracted in Section 2.3 was reverse-transcribed into cDNA with a cDNA synthesis kit (Tsingke, TSK301, Beijing, China). The total reaction system was 20 μL, including 2 μL cDNA, 2 μL 10 M primer F/R, 10 μL 2 × Mix, and 6 μL ddH_2_O. According to the instructions of SYBR^TM^ Green Master Mix (Applied Biosystems™, 00791640, Carlsbad, CA, USA) and the Tm value of the primers, the reaction program was set as follows: pre-denaturation at 95 °C for 2 min, denaturation at 95 °C for 15 s, annealing at 60 °C for 15 s, extension at 72 °C for 1 min, and a total of 40 cycles. The relative quantification of gene expression was performed using the 2^-ΔΔCt^ method.

### 2.5. ELISA

Porcine pulmonary MVECs were divided into groups as described in Section 2.3 and treated with the virus lysate or the control lysate. After incubation in a maintenance medium for 36 h and 60 h, the LPS group and the PRRSV-LPS group were exposed to LPS at a final concentration of 5 μg/mL for 12 h. The culture supernatant was collected from each group and stored at −80 °C for ELISA detection. Total protein was extracted using RIPA lysate (Beyotime, P0013B, Shanghai, China) for Western blot detection.

The frozen cell supernatants were thawed and centrifuged at 12,000 rpm for 20 min. The supernatants were used to measure IL-1α and IL-8 according to the kit instructions (RayBiotech, ELP-IL1a-1, ELP-IL8-1, Atlanta, GA, USA).

### 2.6. Western Blot

The protein samples were separated using SDS-PAGE (Beyotime, P0012AC, Shanghai, China). The stacking gel was run at 70 V, and the separating gel at 120 V. The proteins were transferred to a PVDF membrane (Merk Milipore, IPVH00010, MA, USA) at a constant voltage of 110 V for 80 min. After blocking with 5% skimmed milk solution (Beyotime, P0216, Shanghai, China) for 1 h, the membrane was incubated with β-actin antibody (Proteintech, 66009, Rosemont, IL, USA) at 1:20000 dilution and VCAM-1 antibody (Santa Cruz, sc-18864, Dallas, TX, USA) at a 1:1000 dilution overnight at 4 °C. After washing, a secondary antibody solution at a 1:20,000 dilution was added and incubated at room temperature for 2 h. Finally, a chemiluminescence kit (Beyotime, P0018S, Shanghai, China) was used to visualize the protein bands using a gel imaging instrument (Tanon, Tanon-5200, Shanghai, China).

### 2.7. Transendothelial Electrical Resistance (TEER)

To assess the impact of PRRSV-LPS stimulation on the barrier function of MVECs, cells were seeded in the insert of the Transwell culture system (Biofil, TCS004024, Guangzhou, China). TEER was measured using a Millicell-ERS endothelial resistance instrument (Milipore, MERS00002, MA, USA). After the resistance stabilized, MVECs were divided into groups and treated as described in Section 2.3. At 0, 36, 48, 60, 72, and 96 h post-infection with the HP-PRRSV HN strain, the resistance values were measured, respectively. TEER was calculated according to the following formula.
Teer (Ω·cm^2^) = Resistance value × Transwell membrane area

### 2.8. TEM Assay

MVECs were seeded in the insert of the Transwell plate, and divided into groups, and treated as described in Section 2.7. At 12 h after LPS stimulation, 2 × 10^5^ neutrophils were added to the insert of Transwell plates. After 4 h incubation, the neutrophils in the lower chamber were collected, stained with trypan blue, and counted with a cytometer, and their mobility was calculated according to the following formula.
Mobility = (neutrophil number in the lower chamber/2 × 10^5^) × 100%

### 2.9. Data Analysis

For transcriptome sequencing data, *p* values were corrected by the Benjamini–Hochberg method [16]. The grayscale of protein bands from Western blotting was quantified using Image J software. All data are expressed as “mean ± standard deviation”, and GraphPad Prism 8.0 software was used for data visualization and one-way analysis of variance. A *p* value less than 0.05 was considered statistically significant.

## 3. Results

### 3.1. Cell Characteristics

The CD31^+^ MVECs purified from the primary cell culture of porcine lungs exhibited a polygonal appearance. They were routinely passaged and grew to confluence in about 5 days (Figure 1A). They could be stably propagated up to passage 7 before entering cellular senescence. Immunofluorescence staining of factor VIII was positive, with approximately 97% of cells exhibiting positive staining (Figure 1B,C). Additionally, cryopreserved porcine peripheral blood neutrophils were recovered and assessed using Trypan blue staining, with approximately 98% viability observed. Neutrophil purity was evaluated by Wright stain to be about 95%, and the cells displayed characteristic segmented nuclei, a prominent feature of neutrophils.

### 3.2. Transcriptome Changes of Porcine Pulmonary MVECs

To investigate the response of porcine pulmonary MVECs to HP-PRRSV and/or LPS stimulation, the transcriptome expression was analyzed by RNA-seq. The results showed that HP-PRRSV HN and LPS alone induced the differential expression of 101 (Figure 2A) and 175 (Figure 2B) genes, respectively, and their combined application resulted in the differential expression of 177 genes (Figure 2C). Venn analysis of the differentially expressed genes from individual treatments revealed only 20 genes in common (Figure 2D), indicating significant differences in the functional changes. Further analysis showed that compared to individual stimulation with HP-PRRSV HN or LPS, their combined treatment induced differential expression of 171 (Figure 3A) and 172 (Figure 3B) genes, respectively. Among these, only 21 genes were in common (Figure 3C). These findings suggest that their combined stimulation induces additional functional changes, and their respective roles are distinct.

### 3.3. GO and KEGG Functional Analysis of Differentially Expressed Genes

To understand the characteristics of the functional changes in porcine pulmonary MVECs induced by HP-PRRSV and/or LPS, gene ontology (GO) enrichment analysis was performed on the differentially expressed genes, and the top 20 significantly enriched GO items were selected to generate a histogram. The results revealed that in both individual and combined stimulation, the most significantly enriched genes were mainly associated with biological processes, although their specific items differed significantly. HP-PRRSV infection mainly induced responses to virus, cellular responses to type I interferon, type I interferon signaling pathway, negative regulation of viral gene replication, positive regulation of tyrosine phosphorylation of STAT protein, and positive regulation of JAK-STAT cascade (Figure 4A). The altered genes were involved not only in the viral immune response, such as CCL5, IFIT2, IFIT3, and SOCS3, but also in the inflammatory response, such as VCAM1, TNFSF15, ITGB4, and HES1 (Figure 4B). LPS mainly induced blood vessel development, positive regulation of smooth muscle cell proliferation, positive regulation of smooth muscle cell migration, cellular response to acid chemical, and cellular response to extracellular stimulus (Figure 4C). The altered genes primarily related to the inflammatory response, such as IL6, IL16, CXCL8, CXCL14, TNFSF15, and ITGA7 (Figure 4D). Furthermore, in comparison to HP-PRRSV alone, the biological processes caused by the combined stimulation mainly included connective tissue development, cardiovascular system development, blood vessel remodeling, regulation of signaling receptor activity, receptor ligand activity, and receptor regulator activity (Figure 4E). Their combined stimulation resulted in more changes in genes related to inflammatory responses, including IL-1α, CXCL2, CXCL8, ITGA1, and ITGB8 (Figure 4F), suggesting that LPS can induce a stronger inflammatory response in HP-PRRSV-pretreated MVECs. In comparison to LPS alone, its combined stimulation with HP-PRRSV mainly induced biological processes such as extracellular structure organization, cellular response to extracellular stimulus, extracellular structure organization, and extracellular matrix organization (Figure 4G). A large number of inflammatory genes, such as CCL 5, VCAM-1, ITGB 8, IGTAV, IL-6, and IL17D, were regulated (Figure 4H), indicating that the LPS-induced inflammatory response is more pronounced when MVECs are pretreated with HP-PRRSV.

To clarify the signaling pathway associated with HP-PRRSV-mediated secondary bacterial infection, KEGG analysis was performed on the differentially expressed genes induced by combined stimulation. In comparison to LPS alone, the enriched pathways mainly included the MAPK signaling pathway, TNF signaling pathway, C-type lectin receptor signaling pathway, and IL-17 signaling pathway (Figure 5B). Similarly, compared to HP-PRRSV alone, the enriched pathways primarily comprised the PPAR signaling pathway, MAPK signaling pathway, cytokine–cytokine receptor interaction, and RIG-I-like receptor signaling pathway (Figure 5A). The results show that numerous inflammatory pathways are induced by their combined stimulation.

### 3.4. Validation of Differentially Expressed Genes by qRT-PCR

Given that many differentially expressed genes caused by HP-PRRSV and LPS stimulation, either alone or in combination, are related to immune and inflammatory responses, eight genes that have been well studied in immune and inflammatory responses with definitive roles were selected for qRT-PCR analysis to confirm the RNA-sequencing data. The results showed that the combined stimulation induced significantly upregulated mRNA expression of IL-1α, CXCL8, CXCL2, and ITGB8 compared to PRRSV stimulation alone, and that of ITGAV, CCL5, VCAM1, and TNFAIP3 compared to LPS stimulation alone (Figure 6). These findings were consistent with the RNA-seq results, indicating the reliability of the transcriptome sequencing results.

### 3.5. Changes of Three Differentially Expressed Molecules at the Protein Level

To evaluate the effect of the combined stimulation on the protein level of key inflammatory genes, the production of IL-1α, CXCL8, and VCAM-1 was examined by ELISA and Western blot. There was no detectable IL-1α at 48 h, while its concentration in the PRRSV-LPS group was significantly higher than that in the LPS group at 72 h (*p* < 0.01) (Figure 7A). The concentrations of CXCL8 in the PRRSV-LPS group were significantly higher than those in the PRRSV and LPS groups at 48 h and 72 h (*p* < 0.01) (Figure 7B). The expression of VCAM-1 in the PRRSV-LPS group was significantly higher than that in the LPS group at 48 h and 72 h (*p* < 0.01), and its increase compared to the PRRSV group had no significant difference (*p* > 0.05) (Figure 7C,D).

### 3.6. Reduced TEER of Porcine Pulmonary MVECs by HP-PRRSV and LPS Stimulation

In addition to the expression of inflammatory molecules, disruption of the endothelial barrier is another characteristic of inflammatory response. In this study, the impact of HP-PRSSV and LPS stimulation on the TEER of porcine pulmonary MVECs was evaluated using a Transwell model. The results showed that the TEER reached a plateau stage at 48 h following the seeding of MVECs (Figure 8A), indicating the formation of a confluent monolayer. The stimulation of HP-PRRSV and LPS, either alone or in combination, significantly lowered the TEER compared to the control group (Figure 8B), and the TEER in the PRRSV + LPS group was significantly lower than that in the PRRSV or LPS group. The results suggest that HP-PRRSV HN and LPS exert a synergistic destruction of the endothelial barrier.

### 3.7. Increased TEM of Neutrophils by HP-PRRSV and LPS Stimulation

Given the evidence that stimulation with HN-PRRSV and LPS induces differential expression of numerous inflammatory factors and disrupts the endothelial barrier, the Transwell model was further used to investigate their effect on TEM of neutrophils. The results showed a significant increase of neutrophil TEM in response to the stimulation of PRRSV and/or LPS (*p* < 0.01), and the PRRSV-LPS group exhibited a significantly higher migration rate compared to the PRRSV and LPS group (*p* < 0.01) (Figure 9), suggesting a remarkable synergistic effect of the combined stimulation.

## 4. Discussion

To explore the mechanism underlying the severe lung damage and high mortality rate in piglets caused by HP-PRRSV-mediated secondary bacterial infection, this study analyzed the transcriptome expression, TEER of the cell monolayer, and neutrophil TEM in porcine pulmonary MVECs treated with HP-PRRSV HN strain and LPS. The finding revealed distinct response patterns of porcine pulmonary MVECs to HP-PRRSV and LPS stimulation. HP-PRRSV infection resulted in differential expression of genes related to antiviral immunity and inflammatory response, while LPS stimulation mainly caused differential expression of genes associated with the inflammatory response. Interestingly, their combined stimulation increased the expression of inflammatory factors induced by LPS and exacerbated the functional damage caused by HP-PRRSV infection. Moreover, the combined stimulation significantly intensified the decrease in the TEER of MVECs and synergistically increased the neutrophil TEM. These results suggest that the combined stimulation of HP-PRRSV and LPS has the potential to synergistically enhance the production of inflammatory molecules, induce endothelial barrier dysfunction, and increase neutrophil TEM in porcine pulmonary MVECs. These more severe responses of MVECs may be one of the important mechanisms contributing to the development of severe lung lesions and increased mortality in piglets resulting from HP-PRRSV-mediated secondary bacterial infection.

MVECs have diverse biological functions and serve as crucial points in various physiological and pathological responses, exhibiting specific reactions to different stimuli [17,18]. This study also revealed distinct response patterns of porcine pulmonary MVECs to HP-PRRSV and LPS stimulation. The latter mainly caused differential expression of genes related to inflammatory response, while the former induced differential expression of many inflammatory genes as well as numerous non-specific immune genes, which suggests that porcine pulmonary MVECs not only are involved in the inflammatory lung injury caused by HP-PRRSV infection but also play an important role in the immune response to HP-PRRSV. Moreover, compared with HP-PRRSV or LPS stimulation alone, their combined stimulation resulted in the upregulation of more inflammatory molecules, including interleukins, chemokines, and adhesion molecules, suggesting that the inflammatory response of porcine pulmonary MVECs to HP-PRRSV-mediated secondary bacterial infection is more intense than that to either infection alone. Specifically, HP-PRRSV infection mainly promoted the effects of LPS on the MAPK signaling pathway, TNF signaling pathway, c-type lectin receptors, etc. However, LPS stimulation exacerbated the effects of HP-PRRSV infection on cytokine–cytokine receptor interaction, the RIG-I-like receptor signaling pathway, and so on.

The transcriptome sequencing results were confirmed at the mRNA and protein levels, while different treatment times were selected for the detection between the protein levels and the mRNA levels. In terms of mRNA detection, the treatment times of HP-PRRSV and LPS were 60 h and 24 h, respectively. For protein detection, the HP-PRRSV treatment times were 48 h and 72 h, and the LPS treatment time was 12 h. The results showed that the combined PRRSV-LPS stimulation had stronger effects on the increased production of three inflammatory molecules than either individual stimulation, although their production altered to some extent with the treatment time. Specifically, the secretions of IL-1α and CXCL8 were significantly higher in the PRRSV-LPS group compared to either the PRRSV group or the LPS group, while there was no significant difference between the PRRSV-LPS group and the LPS group at the mRNA level. This inconsistency may be attributed to the longer treatment time of LPS at the protein level, which brought stronger promoting factors for following LPS stimulation. IL-1α is a biologically active precursor released during tissue injury and necrotic cell death [19]. CXCL8, also known as interleukin 8 (IL-8), plays a central role in mediating inflammatory responses by attracting neutrophils to the site of inflammation, releasing vasoactive substances, and causing tissue immune damage [20]. VCAM-1 is involved in processes such as inflammatory response, immune response, and lymphocyte homing [21]. These three inflammatory molecules have been documented to be involved in PRRSV infection [22,23,24]. In this study, porcine pulmonary MVECs were demonstrated to be an important source tissue of inflammatory molecules.

Endothelial barrier dysfunction is an important marker of inflammatory injury in MVECs, and TEER measurement is a common method for evaluating endothelial barrier function [25]. Glycocalyx is the main component of the microvascular endothelial barrier [26]. Our previous study has demonstrated that HP-PRRSV infection leads to the destruction of the endothelial glycocalyx in porcine pulmonary MVECs [13], suggesting a potential impairment of their barrier function. This present study demonstrated that HP-PRRSV infection resulted in a decrease in the TEER of porcine pulmonary microvascular endothelial cell monolayer, although its effect was less pronounced compared to that of LPS stimulation alone. Furthermore, their combined stimulation led to a more substantial decline in TEER, indicating that the endothelial barrier dysfunction was exacerbated.

The neutrophil TEM experiment revealed that while HP-PRRSV HN and LPS stimulation individually increased neutrophil mobility to a similar extent, their combined stimulation significantly augmented it. Both the increased production of inflammatory factors and impaired endothelial barrier function may be the contribution of enhanced neutrophil TEM. During the process of neutrophils crossing MVECs, there is a complex interaction between them, and the function of neutrophils that migrate across the endothelium monolayer may be altered [27]. The combined stimulation of HP-PRRSV and LPS exacerbated the dysfunction of porcine pulmonary MVECs, likely impairing their regulation of neutrophil function. The functional status of neutrophils across impaired endothelial barrier deserves further study.

## 5. Conclusions

This study demonstrated the synergistic effect of the combined stimulation of HP-PRRSV and LPS on the inflammatory injury of porcine pulmonary MVECs. The findings of this research shed light on the underlying mechanism of severe lung injury caused by a secondary bacterial infection following HP-PRRSV infection, with a particular emphasis on the vital role of pulmonary MVECs in HP-PRRSV infection.

## Figures and Tables

**Figure 1 viruses-15-01523-f001:**
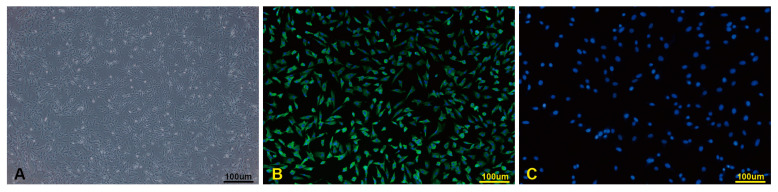
Microscopic morphology of porcine pulmonary MVECs and immunofluorescence staining for factor VIII. (**A**): Microscopic morphology of CD31^+^ MVECs; (**B**): immunofluorescence staining for factor VIII; (**C**): negative control for immunofluorescence for factor VIII. (bar = 100 μm).

**Figure 2 viruses-15-01523-f002:**
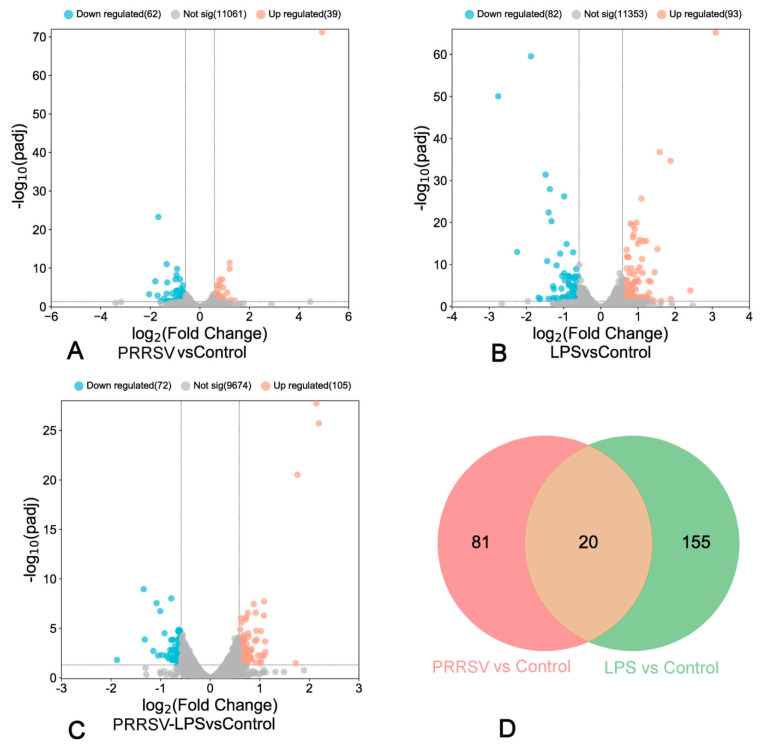
Volcano plot and Venn diagram illustrating differentially expressed genes in porcine pulmonary MVECs induced by HP-PRRSV and/or LPS. (**A**): Volcano plot of the PRRSV group versus control group; (**B**): volcano plot of the LPS group versus control group; (**C**): volcano plot of the PRRSV-LPS group versus control group; (**D**): Venn diagram of the HP-PRRSV group versus control group and the LPS group versus control group.

**Figure 3 viruses-15-01523-f003:**
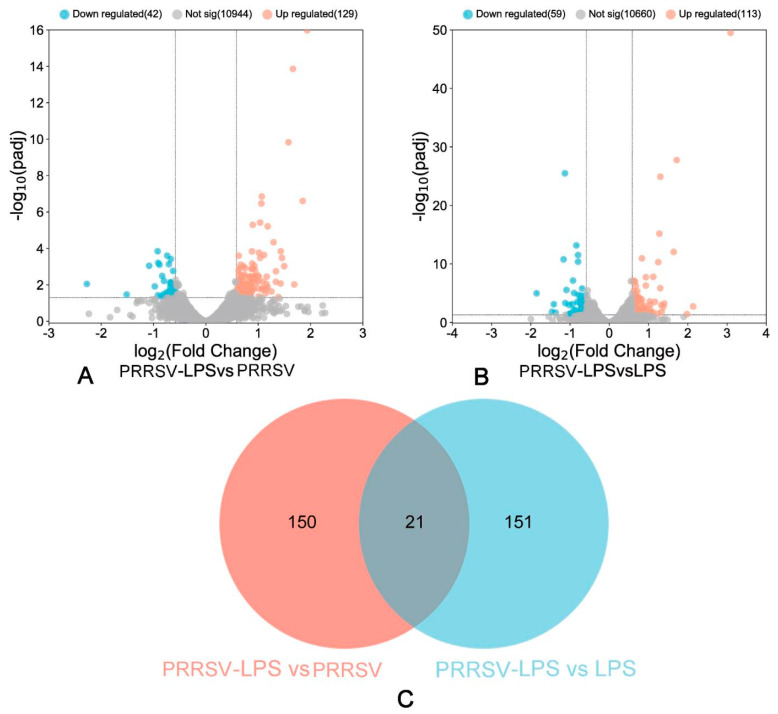
Volcano plot and Venn diagram demonstrating differentially expressed genes induced by combined stimulation compared to individual stimulation. (**A**): Volcano plot of the PRRSV-LPS group versus PRRSV group; (**B**): volcano plot of the PRRSV-LPS group versus LPS group; (**C**): Venn diagram of the PRRSV-LPS group versus PRRSV group and the PRRSV-LPS group versus LPS group.

**Figure 4 viruses-15-01523-f004:**
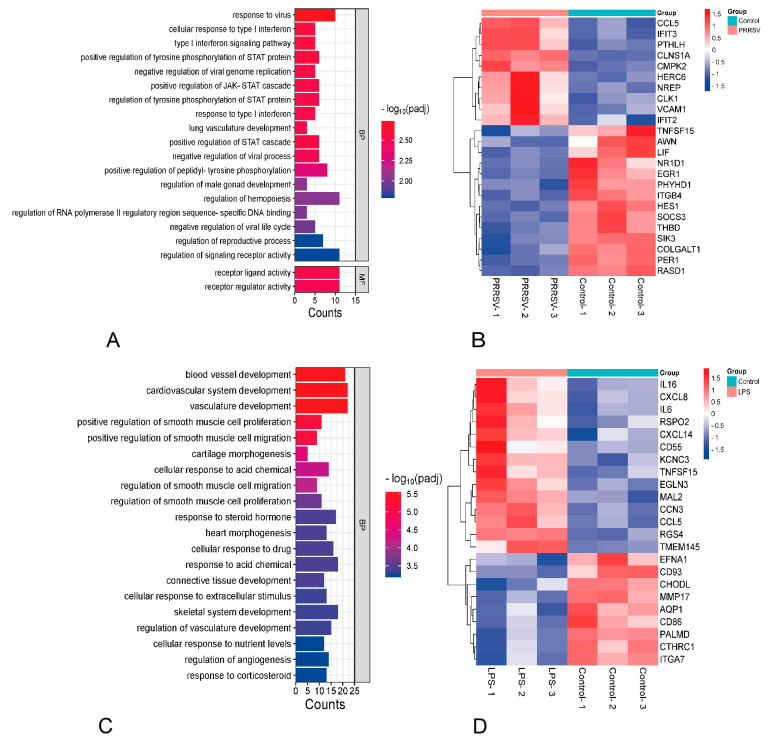
Histogram of GO enrichment analysis and heatmap of primarily differential genes. (**A**,**B**): PRRSV group versus control group, (**C**,**D**): LPS group versus control group, (**E**,**F**): PRRSV-LPS versus PRRSV group, (**G**,**H**): PRRSV-LPS group versus LPS group.

**Figure 5 viruses-15-01523-f005:**
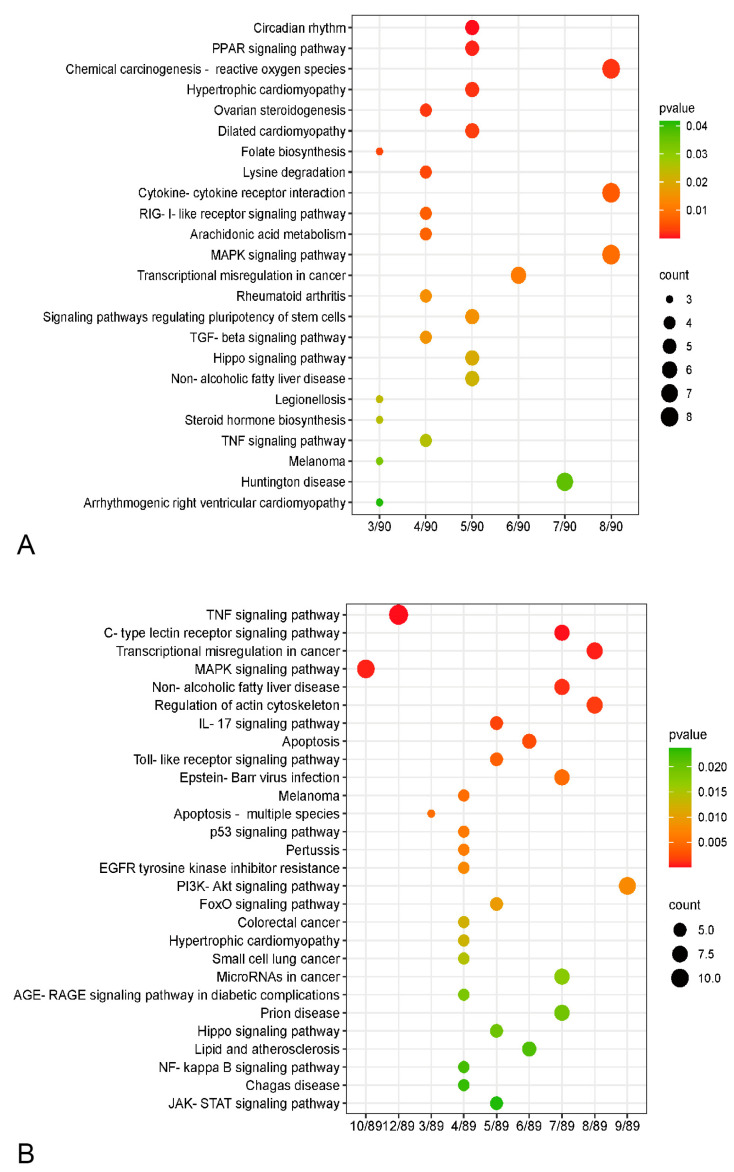
Histogram of KEGG enrichment analysis. (**A**): The PRRSV-LPS group versus the PRRSV group; (**B**): the PRRSV-LPS group versus the LPS group.

**Figure 6 viruses-15-01523-f006:**
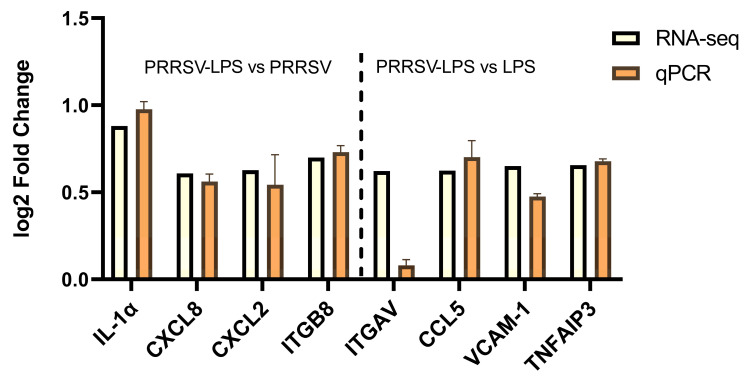
The mRNA expression ratios of representative inflammatory molecules by qRT-PCR and RNA sequencing.

**Figure 7 viruses-15-01523-f007:**
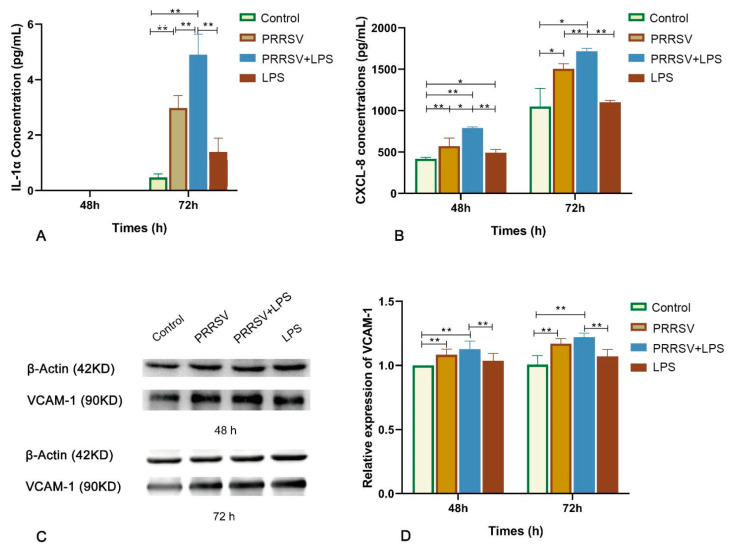
Effect of PRRSV and LPS stimulation on the production of, IL-1α (**A**), CXCL-8 (**B**), and VCAM-1 (**C**,**D**) in porcine pulmonary MVECs at the protein level. * indicates *p* < 0.05; ** indicates *p* < 0.01.

**Figure 8 viruses-15-01523-f008:**
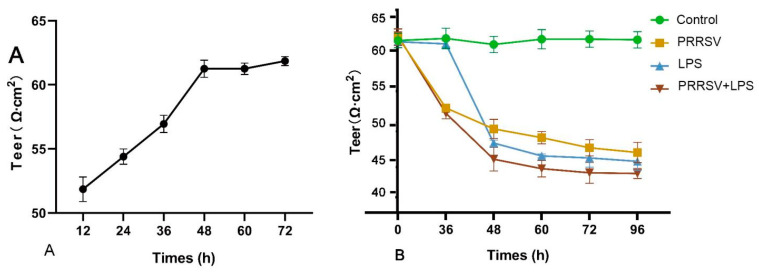
TEER changes of porcine pulmonary MVECs (**A**) induced by effects of HP-PRRSV and LPS stimulation (**B**).

**Figure 9 viruses-15-01523-f009:**
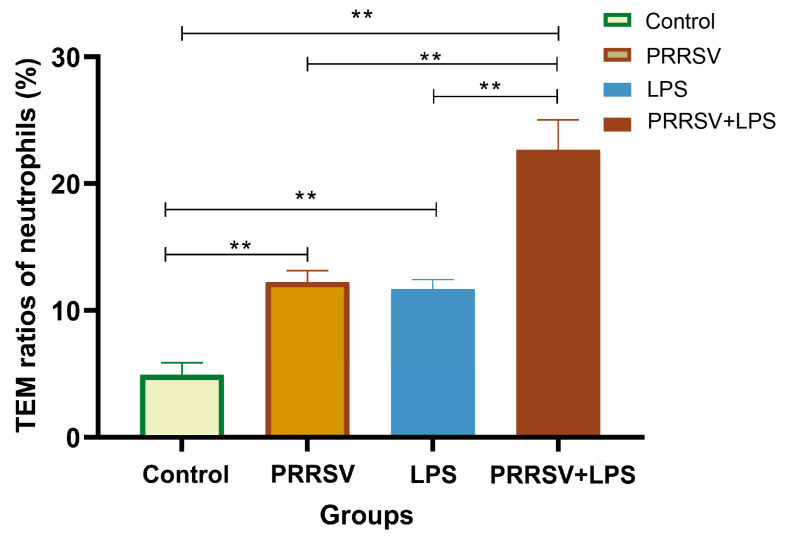
Effect of HP-PRRSV and LPS on neutrophil transendothelial migration. ** indicates *p* < 0.01.

**Table 1 viruses-15-01523-t001:** Gene primer sequence for qRT-PCR detection.

Genes	Primer Sequences (5′-3′)
*IL1A*	F: AAC CTG GAT GAG GCA GTG AAA
	R: AGC ACT CAC AAA CAG TCG GG
*CXCL8*	F: AAT ACG CAT TCC ACA CCT TT
	R: TGT TGT TGT TGC TTC TCA GT
*ITGB8*	F: ACT GGG CCA AAG TGA AGA AAA C
	R: ATC CTC TTG AGC ACA CCA TCC
*ITGAV*	F: CAG CGC GTC TTC GAT GTT TC
	R: CCG GTG AGA AGA CCA GTC AC
*VCAM1*	F: ACG CTT GAC GTG AAA GGA AG
	R: CAC CCC GAT GGC AGG TAT TA
*CCL5*	F: CAT GGC AGC AGT CGT CTT TAT C
	R: AAG TTT GCA CGA GTT CAG GC
*TNFAIP3*	F: ATC CGA CCC CTA CCG TGA C
	R: GGT GCT CTA CAA GGC CTC TC
*CXCL2*	F: GAT GCT AAA CAA GAG CAG TGC C
	R: CCC AGG GGC TAT TTG CTT CTC

## Data Availability

The datasets generated for this study are available on request to the corresponding author.

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
