# Peer review of "Synergistic Effect of Treatment with Highly Pathogenic Porcine Reproductive and Respiratory Syndrome Virus and Lipopolysaccharide on the Inflammatory Response of Porcine Pulmonary Microvascular Endothelial Cells"

_viruses, 2023, doi:10.3390/v15071523_

Round 1
Reviewer 1 Report
Comments and Suggestions for Authors
This is a well written and well presented manuscript examining the effects of PRRSV-2 and LPS exposure for porcine pulmonary microvascular endothelial cells.
The experimental design is fully suitable and robust, and the conclusions are supported by the findings of the study.
The data are clearly presented with useful and well-labelled figures.
Specific comments:
1- The referencing in the text throughout must be updated.
2- Section 2.3: Please indicate how much virus supernatant / lysate was used.
3 - Section 2.6: Please specify what membrane was used (type & supplier).
Comments on the Quality of English LanguageOnly minor editing is required.
Reviewer 2 Report
Comments and Suggestions for Authors
This manuscript briefly described the synergistic effect of the combined stimulation of HPPRRSV and LPS on the inflammatory injury of porcine pulmonary MVECs. This research could help reveal the mechanism underlying severe lung injury caused by secondary bacterial infection following HP-PRRSV infection, with a particular emphasis on the vital role of pulmonary MVECs in HP-PRRSV infection.
This report is helpful for further development of new and safe drug to prevent or treat HP-PRRSV and bacteria co-infection in pig. It is acceptable for publication after the following problems are solved or clarified.
1. There are plenty of grammatical mistakes and verbose expressions in the manuscript, which should be carefully corrected to be succinct.
2. The abstract did not contain any clear data of inflammatory responses caused by co-infection with HP-PRRSV and LPS, which is of importance to reveal their synergistic effect on the co-inoculation of porcine pulmonary microvascular endothelial cells. They should be added in the revised manuscript.
3. The legend color in Fig.9 seems to be wrong between LPS and PRRSV+LPS group, which should be corrected.
Comments on the Quality of English LanguageThere are plenty of grammatical mistakes and verbose expressions in the manuscript, which should be carefully corrected by a native expert to be succinct.
Reviewer 3 Report
Comments and Suggestions for Authors
I reviewed the manuscript entitled “Synergistic effect of treatment with highly pathogenic porcine reproductive and respiratory syndrome virus and lipopolysaccharide on the inflammatory response of porcine pulmonary microvascular endothelial cells”
Overall, I think the manuscript covers an interesting topic. However, considering the methodology used by the authors I consider that more results should be included in the manuscript regarding the antiviral and inflammatory responses pathways. My suggestion to improve the quality of this study would be, the inclusion of a detailed information about the RNA-SEQ analysis conducted in this study. Please, show a detailed comparison about the differential gene expression analysis between different treatments included in this study regarding genes associated with the viral immune response and the inflammatory response. In the current version, there are a limited number of genes. In my opinion more support is needed.
The different steps involved in these responses may be shown using heat maps. Because of the addition of this analysis, different sections in the manuscript have to be updated.
Round 2
Reviewer 3 Report
Comments and Suggestions for Authors
I like to thank the authors for their responses. At this point, I don't have more concerns about this manuscript.